# Epidemiological Characteristics of Cancer Patients Attending at Felege Hiwot Referral Hospital, Northwest Ethiopia

**DOI:** 10.3390/ijerph20065218

**Published:** 2023-03-22

**Authors:** Muluken Azage, Serkalem Zewudie, Martha H. Goedert, Engda G. Hagos

**Affiliations:** 1School of Public Health, College Medicine and Health Sciences, Bahir Dar University, Bahir Dar P.O. Box 79, Ethiopia; 2Department of Oncology, Felege Hiwot Referral Hospital, Amhara Region, Bahir Dar P.O. Box 74, Ethiopia; 3Department of Health Promotion, Social and Behavioral Health, College of Public Health, University of Nebraska Medical Center, Omaha, NE 68198, USA; 4Department of Biology, Colgate University, Hamilton, NY 13346, USA

**Keywords:** cancer hot pots, epidemiology, Amhara region, Ethiopia

## Abstract

Background: Cancer has become a public health problem and a challenge in developing countries, including Ethiopia. There is scanty local data on cancer epidemiology in Amhara region, Ethiopia. Thus, this study aimed to describe epidemiological characteristics of cancer patients attending Felege Hiwot Referral Hospital. Methods: This study was based on a patient cancer registry that took place in Bahir Dar Felege Hiwot Referral Hospital, Amhara Regional State, Ethiopia. It is the main referral hospital in the Amhara region, and serves more than 5 million people. The hospital has units including oncology for follow-up health care services. All confirmed cancer patients attending oncology units from July 2017 to June 2019 were included in the study. Global Moran’s I statistic was employed to assess spatial heterogeneity of cancer cases across districts. Getis–Ord Gi* statistics was performed to identify hot spot districts with high numbers of cancer cases. Results: In a two-year period, a total of 1888 confirmed cancer patients were registered. There was a significant variation of cancer patients between females (60.8% 95%CI 58.5 to 63.0%) and males (39.3% 95%CI 37.0 to 41.5%). The first three most frequent cancer types seen were breast (19.4%) and cervical cancer (12.9%), and lymphoma (15.7%). Breast and cervical cancer and lymphoma were the first three cancers type among women, whereas lymphoma, sarcoma, and lung cancer were the three most common cancer among men. Spatially, cancer cases were non-random in the study area (global Moran’s I = 0.25, z-score = 5.6, *p*-value < 0.001). Bahir Dar city administration (z = 3.93, *p* < 0.001), Mecha (z = 3.49, *p* < 0.001), Adet (z = 3.25, *p* < 0.01), Achefer (z = 3.29, *p* < 0.001), Dangila (z = 3.32, *p* < 0.001), Fogera (z = 2.19, *p* < 0.05), and Dera (z = 2.97, *p* < 0.01) were spatially clustered as hotspot with high numbers of cluster cases. Conclusions: We found that there is a variation in the cancer types with sex. This study provides an insight for further exploration of environmental and occupational exposure related factors for cancer to guide future cancer prevention and control programs. The current study also calls for expansion of cancer registry sites, including in rural areas in the region.

## 1. Background

Cancer is one of the major fatal non-communicable diseases (NCDs) worldwide. It was responsible for an estimated 9.6 million deaths in 2018 [1]. Approximately 70% of deaths from cancer occur in low- and middle-income countries [1,2,3]. In Africa, the incidence and mortality from cancer is rising rapidly in the continent, with estimates of 847,000 new cases and 591,000 cancer deaths in 2012 [4]. It has been projected that by 2030, the number of cancer cases and death in Africa is expected to grow rapidly by at least 70% [4].

Studies show that the increase in cancer incidence is due to population growth; ageing; change of lifestyle habits such as smoking, physical inactivity, unhealthy diet, and high calorie intake; and certain infectious agents [4,5]. The United Nations estimated that the population of Africa between 2010 and 2030 is projected to increase by 60% overall and 90% of those are 60 years and older, the age at which cancer most frequently occurs [4].

Recent studies show that the incidence of cancer is rapidly increasing in Ethiopia [6,7]. It is one of the top leading causes of death in the adult population and it accounts for about 5.8% of the total national mortality [8]. Furthermore, a study conducted in Ethiopia in 2015 revealed that the incidence of cancer cases was more than 64,000, with a significant difference between males (21,563 with 95%CI, 17,416 to 25,660) and females (42,722 with 95%CI, 37,412 to 48,040) [9]. This study also reported that the commonly occurring cancers in Ethiopia were breast, cervical, and colorectal cancer, non-Hodgkin’s lymphoma, leukemia, cancers of the prostate, thyroid, lung, stomach, and liver cancer [9]. 

Although most cancers are treatable, the psychosocial impact of cancer on the individual, family, and society is extremely high, particularly in developing countries where limited cancer treatment and counseling centers are available [10,11]. Cancer causes both physical and psychological suffering, as well as causing negative social and spiritual experiences in the lives of the patient and their families [10]. A cancer diagnosis can create extreme disruption in the life of almost any individual and a threat to one’s general sense of security and orderliness in life. Many people, including cancer patients, have a deep fear that any cancer represents pain, suffering, and is fatal. Cancer interventions and treatments (e.g., surgery, chemotherapy, radiotherapy, and hormonal therapy) also seriously affect body image [12,13]. Studies showed that cancer patients experienced a variety of mental health problems (anxiety and depression) and emotional stress, which can cause challenges in everyday life, such as not being able to work, financial problems, and a lack of social support [14,15]. 

Ethiopia is one of the developing countries with few cancer treatment centers located in the capital city (Addis Ababa) and regional metropolitan cities. The national cancer incidence was estimated based on single-city population-based registry data (Addis Ababa) supplemented with secondary data from six regional cancer treatment centers [9]. Although the Amhara region is the second largest region in the country, only a few cancer treatment centers established recently (three in number), exist in the region. Likewise, in the Amhara region, there is scanty local data on cancer epidemiology due to the lack of a surveillance system and limited healthcare service availability and accessibility. Such conditions contribute to the disease progression in patients, meaning it can reach an advanced stage that is not preventable due to the lack of early detection and treatment. Early detection and treatment of cancer is known to greatly reduce the burden of cancers and improve outcomes of cancer patients. The government of Ethiopia has developed a national cancer control plan that has a range of preventive intervention strategies [16]. Hence, understanding cancer epidemiology is paramount to devising new strategies and strengthening existing strategies to prevent and control cancer. Therefore, the objective of the study was to describe epidemiological characteristics of cancer patients attending Felege Hiwot Referral Hospital, Amhara region, northwest Ethiopia. 

## 2. Methods

Study area and population: This study was conducted in Felege Hiwot Referral Hospital, based on a patient cancer registry. The hospital is located in the capital city of Amhara Regional State, Bahir Dar, which is located at 11°36′ latitude N and 37°23′ longitude E in the north–west Ethiopia with an altitude of 1800 m. According to the Amhara Bureau of Finance and Economic Development 2016 report, the estimated population of the Amhara region based on the 2007 national census is ~21 million, of which males are 49.9%, and 40.1%, 55.7%, and 4.2% are in the age groups ≤14, 15–64, and ≥65 years of age, respectively [17]. Felege Hiwot Referral Hospital is the main referral hospital in the Amhara region of Ethiopia, and covers a widespread population of 5 million people. The hospital has surgery, medical, pediatrics, obstetrics and gynecology, psychiatry, dental, oncology, and orthopedics units with both out-patient and in-patient departments and follow-up departments. The oncology unit was established in 2017 and started service for in- and out-patients in July, 2017. There is a limited number of beds for in-patents (19 beds). The center is running with 2 oncologists, 4 general practitioners, 10 pharmacists, and 10 nurses. The hospital is equipped with different cancer diagnosis tools, such as tissue biopsy and imaging, to identify the type of cancer. There is a dedicated pharmacy department for the safe compounding of oncology drugs.

Expansion of cancer diagnostic and treatment centers is one of the strategies stated under the Ethiopian national cancer control plan, which was developed in 2015. Its mission is to build a health care system that is equipped, staffed, trained, and empowered to provide a full range of cancer prevention, screening, diagnostic, treatment, and care options to cancer patients in Ethiopia. The goal of the national plan is to reduce cancer incidence and mortality in Ethiopia by 15% by 2020. The national cancer control plan set an objective to avail access to cancer diagnosis and treatment to 30% of new cancer patients by 2020 [16]. With the above consideration, Felege Hiwot hospital has started cancer diagnosis and treatment services to contribute to the targets of the national cancer control plan.

All confirmed cancer patients who came to out-patient or in-patient departments of oncology units from July 2017 to June 2019 were included in this study.

Data source and collection: All patients suspected for cancer from different units of the hospital were referred to the cancer unit for clinical assessment and diagnostic investigations. The diagnosis of cancer was confirmed through pathologic examination of a biopsy specimen, laboratory and imaging modalities (a computerized tomography (CT) scan and magnetic resonance imaging (MRI)). A pathological test was used for most of the suspected cases of cancer, whereas a computerized tomography (CT) scan and MRI were used as a diagnostic tool for some cancer patients. Confirmed cancer patients were referred to the oncology unit for treatment and management of the disease. Patient history including socio-demographic characteristics, such as age, sex, and location, were registered during the first day of arrival at the oncology unit. The primary site (topography) and histology (morphology) of the malignancies were identified according to the International Classification of Diseases for Oncology, published by the World Health Organization (WHO) in 2000 [18]. Data were extracted from the registry using data collection checklist format, which contains demographic characteristics of patients such as age, sex, place of residence, cancer type, diagnosis, and treatment type, and pathology result (Appendix A). Data were collected by trained nurses and extracted data were cross checked with patient documents by a physician to assure the accuracy of the data.

The projected population data based on the 2007 National Population and Housing Census results of Ethiopia in a district for the year 2017/2018 and 2018/19 were obtained from the Bureau of Finance and Economic Development, Amhara Regional National State [19]. The projected population was used as the known underlying population at risk to calculate the incidence rate of cancer for each district. Cancer cases (*n* = 82) whose underlying population at risk and came from other regions (Benishangul-Gumuz and Oromia Regions) are unknown, and were excluded for incidence rate calculation. 

Data analysis: Data were entered using Stata version 14 for analysis. Descriptive statistics such as mean, proportion, and standard deviation were used to describe data. District locations as clusters were created to explore the spatial analysis of cancer cases. Incidence rate was calculated for age, sex, and each district. The district administration of latitude and longitude were taken as the centroid locations. A total of 52 clusters with cancer cases were created using ArcGIS version 10.8 software. Spatial autocorrelation (global Moran’s I) statistic was employed to assess spatial heterogeneity for cancer cases. The global Moran’s I values close to −1 indicate that poor practice is dispersed, close to +1 indicates clustered, and zero indicates randomly distributed cancer cases [20]. Hot spot analysis (Getis–Ord Gi* statistics) z-scores and significant *p*-values gave the features with either hot spot or cold spot values for the services spatially. ArcGIS 10.1 was used to map cancer patients by location. 

Ethical consideration: Ethical approval was obtained from the Institutional Review Board of College of Medicine and Health Sciences (CMHS/IRB 03-008) (Appendix A), Bahir Dar University. Since we used a retrospective study of medical records, the IRB of CMHS waived the requirement for informed consent. A formal letter was written to Felege Hiwot Referral Hospital to obtain permission for extraction of data from patient documents. Data were extracted anonymously. 

## 3. Results

During a two-year period from July 2017 to June 2019, 1888 cancer patients visited the oncology unit at Felege Hiwot Referral Hospital. More than 60.8% (1148) were females and there was significant variation in cancer patients between females (60.8% 95%CI 58.5 to 63.0%) and males (39.2% 95%CI 37.0 to 41.5%). Nearly half of the cancer patients (48.1%) were found in the age range between 35 and 54 years. A significantly higher proportion of female cancer patients, compared to males, were in the age range between 25 and 34 years, 35 and 44 years, and 45 and 54 years (Table 1).

The incidence of cancer per 100,000 population was 19.85 and there was a higher incidence among females (23.91 per 100,000 population) than males (15.69 per 100,000 population). Across all age groups, the incidence of cancer among female was higher compared to males. The highest incidence of cancer (106.51 per 100,000 population) occurred in the age range between 55 and 64 years. The incidence of cancer in the age between 55–64 years was the highest in both sexes (male vs. female, 98.7 vs. 114.17 per 100,000 population, respectively) and the lowest in the under 15 years of age group (male vs. female, 1.30 vs. 1.36 per 100,000 population, respectively) (Figure 1). The highest incidence of cancer was located in Bahir Dar City (95.37 per 100,000), Dangila (46.14 per 100,000), and Gonjii Kollela districts (39.82 per 100,000 population.) The incidence of cases in 2017/18 and 2018/19 was 6.09 and 13.69 per 100,000 population, respectively (Appendix A).

The data show that almost 50% of the patients are in the age range between 35 and 54 years. The first six most frequent cancer types seen at the oncology unit were breast (19.4%) and cervical cancer (12.9%), lymphoma (15.7%), sarcoma (7.3%), colorectal (6.4%), and lung cancer (6.0%) (Table 2 and Figure 2). 

Breast and cervical cancer, lymphoma, ovarian cancer, and sarcoma were the first five most common cancer types among females, whereas lymphoma, sarcoma, lung cancer, colorectal, and leukemia were the first five most common cancers among males (Table 2). We also found that lymphomas (25.5%), Wilms’ tumor (25.2%), and sarcoma (21.1%) were the first three leading types of cancer among children under 14 years of age (Figure 3). More than 96% of cancer patients came from 56 districts, of which the highest (17.7%) came from Bahir Dar city administration. Nearly two-thirds of the cases came from 15 districts (Bahir Dar city administration, Mecha, Adet, Achefer, Dangila, Fogera, Dera, Jabitenan, Gonj Kollela, Buire Womeberma, Este, Hullet Ejuensie, Bahir Dar Zuriya, and LivoKemekem districts (Figure 4). 

Spatially, cancer cases were non-random in the study area (global Moran’s I = 0.25, z-score = 5.6, *p*-value < 0.001.) In the spatial hot spot analysis, Bahir Dar city administration (z = 3.93, *p* < 0.001), Mecha (z = 3.49, *p* < 0.001), Adet (z = 3.25, *p* < 0.01), Achefer (z = 3.29, *p* < 0.001), Dangila (z = 3.32, *p* < 0.001), Fogera (z = 2.19, *p* < 0.05), and Dera (z = 2.97, *p* < 0.01) were spatially clustered (hot spots) with higher numbers of cancer cases (Figure 5).

## 4. Discussion

This study provides scientific information on the epidemiological characteristics of cancer patients, which can be used for public health planners to understand the extent of the problem, and that can be used for resource allocation and to strengthen the prevention strategies against cancer. The International Agency for Research on Cancer’s GLOBOCAN projects that the number of African cancer cases and deaths are expected to grow approximately 70% by 2030 [4]. Cancer is a leading cause of death in Ethiopia’s adult population and accounts for 5.8% of the total national mortality [8]. Over forty-seven percent of the cancer patients (47.7%) were between 35 and 54 years of age in this study. The highest incidence of cancer occurred in the age range between 55 and 64 years. The age-standardized rates (ASR) for the established anatomical sites vary from previous studies in Ethiopia [9]. This study revealed that the first three leading cancers were breast and cervical cancer, and lymphoma, constituting 48% of all cancers. The findings of the most common types of cancer in this study is consistent with previous research in Ethiopia [9]. Moreover, the highest incidence in the current study occurred below 65 years of age, whereas cancer incidence was the highest in the older than 65 years of age cohort in the previous study [9]. Low detection rates and underreporting of cancer patients due to the absence of screening in Ethiopia could influence the observed disparity, and the difference needs further study.

In this study, we found that females were more affected than males, which is consistent with population-based cancer registries in Ethiopia [9,21] and studies performed globally [1]. There is no clear evidence that underlies the sex disparity in cancer risk. However, the possible explanations for sex disparity documented in different studies are attributed to genetic factors, sex-specific hormones, and differential exposure to risk factors such as lifestyle and environmental factors [22,23,24]. 

The most common cancer in men in the current study was lymphoma followed by sarcoma and lung cancer, whereas in the previous study, the most common was colorectal cancer followed by non-Hodgkin’s lymphoma and prostate cancer [9]. The observed disparity could be due to the difference in exposure to environmental factors and availability of diagnosis services in health care facilities. For instance, the current study used data from a single-institutional-based study, whereas the previous study used data from five different regions and one city administration. 

Lymphoma was also documented as a common type of cancer in developing countries by the World Health Organization [25]. Systematic review findings revealed that environmental exposures such as smoking [26,27] and pesticides [28,29] were documented as risk factors. Massive local charcoal production is a core business, and is produced for sale for big cities of Ethiopia such as Addis Ababa and Mekele. There is extensive use of pesticides in agricultural activities by farmers, which are common in the surrounding area. According to the economic and development office, more than 10 million bags of charcoal are sent to the big cities every year in the surrounding three districts (Mecha and Achefer districts from West Gojjam zone and Fagita Lecoma district from Awi zone). Further study is recommended on exploring the risk factors of lymphoma and cervical cancers in the study area.

In women, the most common cancer was breast cancer, followed by cervical cancer and lymphoma. The above-mentioned cancer types were documented as common types of cancer in a previous population-based cancer registry study performed in Ethiopia [9]. Breast cancer, according to Memirie and colleagues (2018), is the most common cancer in Ethiopia, accounting for 33% of all cancers. These data are consistent with our current finding that breast cancer is among the most common cancer types in women. It accounts for 28% of all cancers, followed by cervical cancer at 21% and lymphoma (7%). Evidence shows that breast cancer remains the most common type of cancer in women globally [3,30]. Furthermore, recent findings reveal that incidence rates of breast cancer are increasing in many low- and middle-income countries, even though the absolute rates are lower in low- and middle-income countries compared to high-income countries due to increased mortality rates [31,32,33]. Although breast [33] and cervical cancer [34,35] are comparable with female cancers in low- and middle-income countries, the high percent of lymphoma needs further study. Considerations may include the sociodemographic index (SDI) for public health surveillance, which addresses demographic and epidemiological perspectives, including occupation.

Another cancer with higher incidence is a gestational trophoblastic neoplasia (GTN). We found 17 cases, which accounts for 4.6% of the female cancer diagnoses in this study. GTN is a rare reproductive cancer and its incidence ranges from one to three per 1000 pregnancies and varies country to country [36]. The variation of incidence might be due to differences in prevalence, discrepancies between hospital-based and population-based data, or disparity in the availability of laboratories to diagnose the diseases. The risk factors related to GTN in the current study need further investigation. 

Studying the geographical pattern of cancer can also prove where clusters of cancer cases are occurring for better countermeasures against the disease, giving direction and coordination to control strategies. This study revealed that cancer cases were non-random in the study area. High numbers of cancer cases were clustered in Bahir Dar city administration, Mecha, Adet, Achefer, Dangila, Fogera, and Dera. The possible spatial variation might be explained by the difference in exposure to risk factors for cancer. Investigating risk factors of cancer types in the surrounding districts are an area of research. Populations within these districts have very specific work exposures, including charcoal production, agriculture activities supported with weedicides and pesticides, khat production, and the indoor use of organic materials such as for cooking. Occupational and household practices and exposures need to be further explored in relation to cancer co-factors. 

Investigating the geographical pattern of cancer using global Moran’s scores and hot spot analysis was an advantage in this study to help identify where clusters of cancer cases occurred for better countermeasures against the disease. The sample size, although small, presents premiere data, from the onset of oncology data collection, beginning in July 2017. This two-year data analysis of new-case cancer patients points to significant differences compared to projected national and global data, related to patients’ age and sex differences. Underlying factors that may impact regional new-case cancer incidence are considered by the researchers, along with background information on the environmental, occupational, and lifestyle factors that may impact these patients presenting to the facility for hallmark cancer diagnoses. 

This study’s limitation is that data were only collected from a single health facility for a two year period; therefore, our data could not represent all cancers in the studied districts, and the incidence of cancer across sociodemographic characteristics (age, sex, and districts) may be underestimated for the population at risk during the study period. The cancer cases attending the health facility might be, more commonly, people with better health seeking behavior and socio-economic status. Moreover, some of the cancer cases use other health facilities due to limited access to cancer treatment. For example, in 2019, Ethiopia had only one functioning cobalt teletherapy machine (radiotherapy center) in Addis Ababa, serving more than 100 million inhabitants [37]. 

For the first time, we describe, in detail, the epidemiologic characteristics of cancer patients enrolled at Felege Hiwot Hospital using spatial analysis, providing policymakers and healthcare providers with critical information for patient management and regional cancer response planning. Existing cancer treatment centers are not equitable, as many cancer patients living in rural districts travel long distances to access services. The study also helps the clinical practitioner to plan counseling interventions in line with clinical interventions by nearby health facilities for rural cancer patients. The identified districts with a high proportion of cancer cases should receive priority attention from public health planners for the allocation of scarce resources and to devise interventions. The promotion of a healthy lifestyle; awareness creation; early diagnosis and treatment of cancer; and the further identification of risk factors related to genetics, environment, and occupation are the areas of further research and intervention.

## 5. Conclusions

Cancer has become a public health problem and challenge in Ethiopia, including the region in this study. Lymphoma, breast cancer, and cervical cancer were the common types of cancer registered in Felege Hiwot hospital. There is variation in the cancer types between sex and districts. This study provides an insight for further exploration of environmental and occupational-exposure-related factors for cancer to guide future cancer prevention and control programs. The current study calls for an expansion of cancer registry sites, including in rural areas, in order to have full epidemiological pictures of cancer incidences in the region.

### 5.1. What Is Already Know on This Topic

There is variation in the cancer types between sex and districts;Hot spot districts with cancer cases were identified

### 5.2. What This Study Adds

The most common cancer types for women and men were identified in the study area;Hot spot districts with cancer cases were identified using spatial analysis, which provides an insight for further exploration of exposure-related factors for cancer, in order to guide future cancer prevention and control programs

## Figures and Tables

**Figure 1 ijerph-20-05218-f001:**
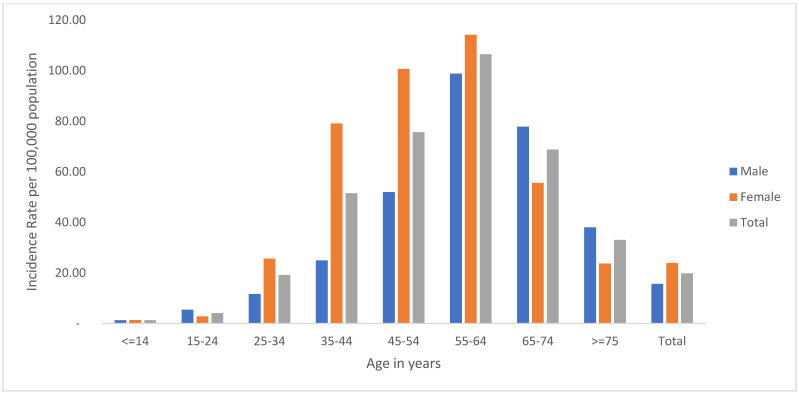
Incidence of cancer per 100,000 population by sex distribution among patients attending oncology department at Felege Hiwot Hospital, Amhara region, 2019.

**Figure 2 ijerph-20-05218-f002:**
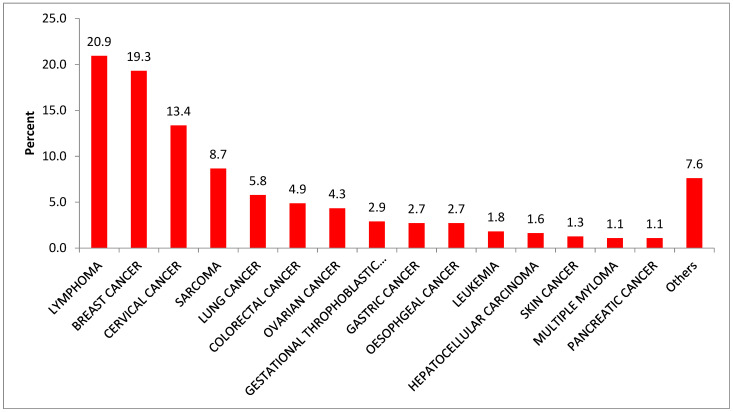
Proportion of cancer types among patients attending at oncology unit, Felege Hiwot Referral Hospital, Amhara region, Ethiopia, 2019. (Other cancers are: anal, Kaposi’s sarcoma, testicular, Wilms’ tumor, neuroblastoma, cancer of unknown primary site, nasopharyngeal, thyroid, vulvar, prostate, salivary gland, small bowel, cholangiocarcinoma, parotid, tonsillar, and uterine cancer.)

**Figure 3 ijerph-20-05218-f003:**
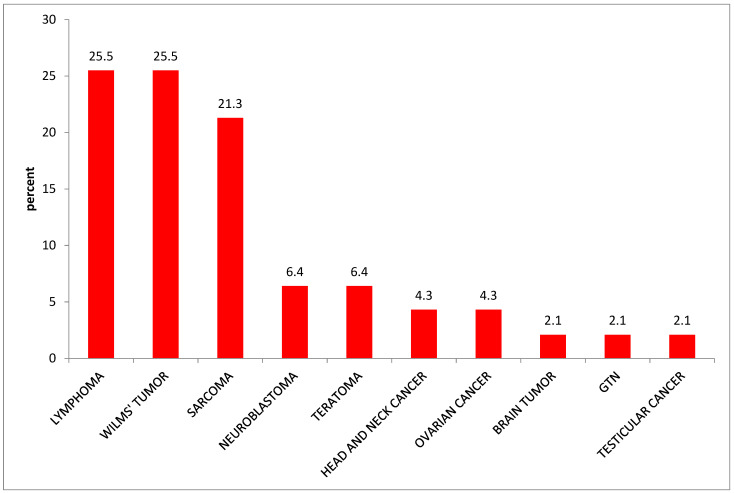
Proportion of cancer type among children 14 and below years of age attending at oncology Unit Felege Hiwot Referral Hospital, Amhara region, Ethiopia, 2019. (GTN = gestational trophoblastic neoplasia).

**Figure 4 ijerph-20-05218-f004:**
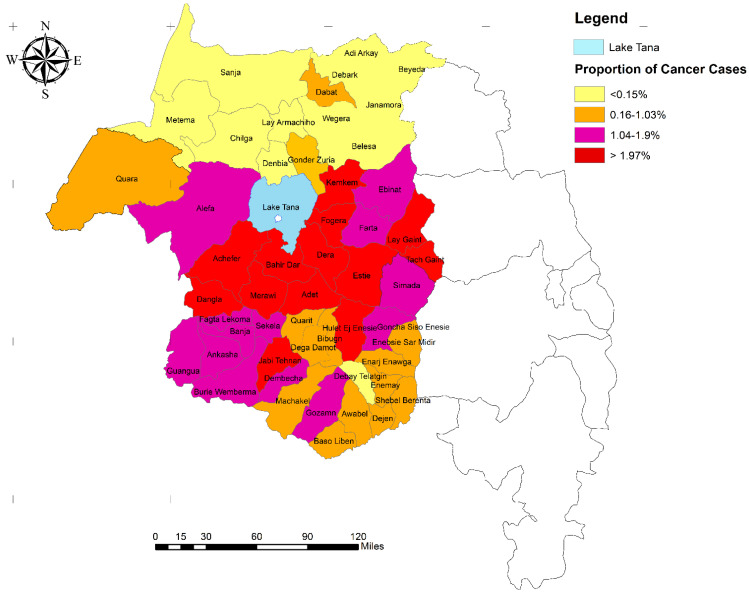
Geographical distribution cancer cases by district, Amhara region, Ethiopia, 2019.

**Figure 5 ijerph-20-05218-f005:**
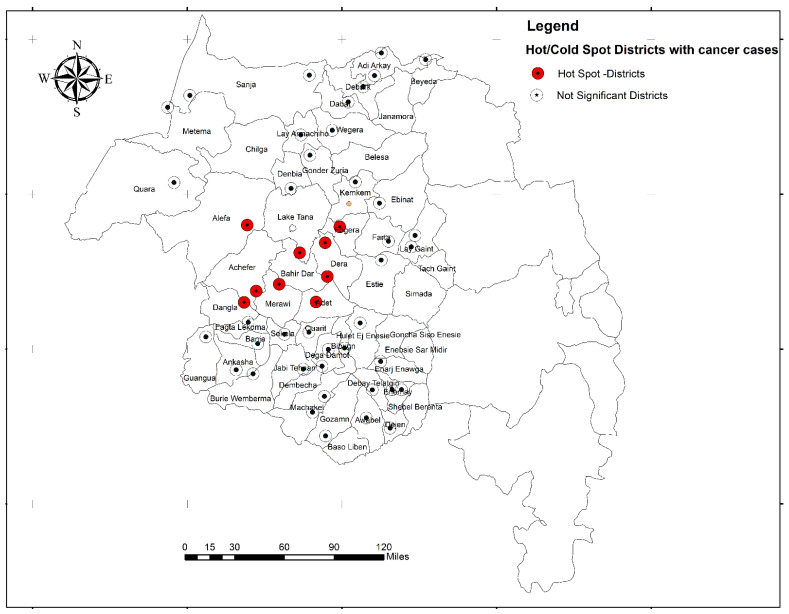
Hot spot districts with high number of cancer cases, Amhara region, Ethiopia, 2019.

**Table 1 ijerph-20-05218-t001:** Age and sex distribution of cancer patients attending at oncology department of Felege Hiwot Hospital, Amhara region, Ethiopia, 2019.

Age	Frequency		Proportion (95%CI)		Total n (Proportion with 95%CI)
	Female	Male	Female	Male	
≤14	22	25	1.9 (1.3–2.9)	3.4 (2.3–4.9)	47 (2.5, 1.9–3.3)
15–24	36	58	3.1 (2.3–4.3)	7.8 (6.1–9.9)	94 (5.0, 4.1–6.1)
25–34	178	72	15.5 (13.5–17.7)	9.7 (7.8–12.1)	250 (13.2, 11.8–14.8)
35–44	344	114	30.0 (27.4–32.7)	15.4 (13.0–18.2)	458 (24.3, 22.4–26.2)
45–54	293	158	25.5 (23.1–28.1)	21.3 (18.5–24.4)	451 (23.9, 22.0–25.9)
55–64	219	190	19.1 (16.9–21.5)	25.6 (22.6–28.9)	409 (21.7, 19.9–23.7)
65–74	46	97	4.0 (3.0–5.3)	13.1 (10.8–15.7)	143 (7.6, 6.5–8.9)
≥75	10	27	0.9 (0.5–1.6)	3.6 (2.5–5.3)	37 (2.0, 1.4–2.7)
Total	1148	741	60.8 (58.5–63.0)	39.2 (37.0–41.5)	1889 (100)

**Table 2 ijerph-20-05218-t002:** Common cancer type by sex attending at oncology department at Felege Hiwot Hospital, Amhara region, Ethiopia, 2019.

Female				Male		
	Cancer Type	No	%	Cancer Type	No	%
1.	BREAST	326	28.4	LYMPHOMA	218	29.4
2.	CERVICAL	242	21.1	SARCOMA	77	10.4
3.	LYMPHOMA	78	6.8	LUNG	66	8.9
4.	OVARIAN	72	6.3	COLORECTAL	61	8.23
5.	SARCOMA	61	5.3	LEUKEMIA	59	8.0
6.	COLORECTAL	60	5.2	HEPATOCELLULAR CARCINOMA	46	6.2
7.	GESTATIONAL THROPHOBLASTIC NEUPLASM	57	5.0	BREAST	41	5.5
8.	LUNG	47	4.1	OESOPHGEAL	26	3.5
9.	GASTRIC	33	2.9	OROPHARYNGEAL	21	2.8
10.	ESOPHGEAL	33	2.9	GASTRIC	19	2.6
11.	HEPATOCELLULAR CARCINOMA	29	2.5	TESTICULAR	14	1.9
12.	HEAD AND NECK	19	1.7	PANCREATIC	12	1.6
13.	LEUKEMIA	10	0.9	HEAD AND NECK	11	1.5
14.	PANCREATIC	10	0.9	SKIN	11	1.5
15.	VULVAR	7	0.6	WILLM’S TUMOR	8	1.08
16.	OROPHARYNGEAL	7	0.6	NASOPHARENGEAL	7	0.94
17.	ANAL	6	0.5	MULTIPLE MYLOMA	6	0.81
18.	CHRONIC MYELOID LEUKAEMIA	6	0.6	CANCER OF UNKNOWN PRIMARY SITE	4	0.5
19.	CANCER OF UNKNOWN PRIMARY SITE	6	0.5	SMALL BOWEL	4	0.5
20.	SMALL BOWEL	6	0.5	PROSTATE	4	0.5
21.	MULTIPLE MYLOMA	5	0.4	ANAL	3	0.4
22.	NASOPHARENGEAL	4	0.4	NEUROBLASTOMA	3	0.4
23.	WILMS’ TUMOR	4	0.4	ACUTE LYMPHOCYTIC LEUKEMIA	2	0.27
24.	TERATOMA	4	0.4	BRAIN TUMOR	2	0.27
25.	KAPOSIS SARCOMA	3	0.3	CHRONIC MYELOID LEUKAEMIA	2	0.27
26.	SKIN	3	0.3	CHOLANGOCARCINOMA	2	0.27
27.	BLADDER	2	0.2	LIPOSARCOMA	2	0.27
28.	CHOLANGOCARCINOMA	2	0.2	TERATOMA	2	0.27
29.	OTHER CANCER	6	1	OTHER CANCER	7	1.0

## Data Availability

Due to third-party restrictions, data are available from Felege Hiwot Hospital. Interested researchers may contact the Medical Director of the hospital in order to access the data: Yesewbelay Minale, Medical director Felege Hiwot Hospital Medical Director Mobile phone: +251921576519 Office phone: +251582264412 Bahir Dar, Ethiopia.

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
