# Peer review of "Epidemiological Characteristics of Cancer Patients Attending at Felege Hiwot Referral Hospital, Northwest Ethiopia"

_ijerph, 2023, doi:10.3390/ijerph20065218_

Round 1
Reviewer 1 Report
Dear Authors,
Thank you for the opportunity to review this paper. In my opinion, the substantive value of the publication submitted for evaluation is low.
Comments and Suggestions for Authors:
Main Comments:
In my opinion, the main and very fundamental weakness of this study is the planned analyzes showing the results of this project.
I do not know what the exact results are presented in Table 1, to what values 95% is calculated. confidence interval and what does the Total column represent?
The Total column does not add up properly, neither do the values in parentheses in this column - they do not add up to 100%, etc.
In addition, are you sure the confidence intervals from table 1 are correctly calculated - e.g. for a value of 1.9 should the confidence interval be 33.1-75.1?, etc.
Figure 1 repeats the results from Table 1 - this should not be the case.
The significance of gender differences (female-male) should be statistically tested since the authors compare frequencies between males and females.
The number of cancer cases should be related to the number of people with given socio-economic characteristics (e.g. age, sex, place of residence).
Authors should seriously consider showing incidence rates rather than numbers (all raw data).
The assumptions of the authors are very good, but the way and scope of presenting the results differs from the standards.
Minor remarks:
Very important information regarding the presented studies, e.g. Institutional Review Board Statement: ?? Informed Consent Statement: ?? Data Availability Statement: ??, has not been properly supplemented by the authors. Applies to section from line 295 - please complete.
The list of literature requires editing, standardization and adaptation to the standard of the journal.
Keywords should be different than in the title - please change.
Numerous editorial errors, commas instead of dots, missing dots, in tab 1 - additions are necessary, e.g. in what units is age, etc. - please be more careful in the future.
Thank you.
Reviewer 2 Report
Excellent study.
It would be nice if you can provide a pictorial presentation of cancers in the other states or districts of Ethiopia and discuss how they compare with the present study.
You may also present a brief paragraph or a picture of the composition of the Cancer Registry at this hospital as well as the format in which the data was abstracted from case notes.
A few spellings may be corrected: e.g., Wilm's in Fig 3.
Also the abbreviation, GTN may be used for gestational trophoblastic tumours, in the same figure.
Is ti possible to know what type of lymphomas are common? e.g., Burkitts in paediatric, Hodgkin's, etc?
Reviewer 3 Report
Dear Authors,
congratulations on your valuable work. Please, find here below some suggestions that I hope will further improve the quality of your paper.
1. Line 59: Do you mind "incident" or "incidence"? Please, specify.
2. Please, be careful with repetitions. For example, in the introduction you used the word "cancer" too often. You could consider to reformulate some phrases and to use synonims, in order to improve the readability of the text.
3. In my opinion, it could be useful to describe, in the introduction, the facilities dedicated to cancer treatment available in your Region. Is the Felege Hiwot Referral Hospital the only oncological hospital in the Region? Which is the situation in the whole country? Furthermore, I think it could be useful to better explain how your hospital works: is there a dedicated pharmacy department for the safe compounding of oncology drugs? Do you manage in-patients and out-patients? Please, add some further details about your centre in order to provide to the readers a more comprehensive picture of your own reality.
4. Line 119: should 'data analysis' be stated in bold?
5. Please, improve the quality of figures 1, 2, 3 and 5. In particular, regarding figure 2, you could merge all tumour types with a frequency of less than 1%, so that you have an easier-to-read graph.
6. Please choose how many decimals (one or two) to insert after the decimal point and keep the same style throughout the paper.
Reviewer 4 Report
Dear authors, thank you for your interesting contribution. Here are my suggestions.
INTRODUCTION
1. Background is well-presented but lacks some details. What about body image in cancer survivors? (see Sebri et al., 2022)
2. Again, please add more details about emotional issues in cancer survivors (see: Durosini et al., 2022)
References
- Durosini, I., Triberti, S., Savioni, L., Sebri, V., & Pravettoni, G. (2022). The Role of Emotion-Related Abilities in the Quality of Life of Breast Cancer Survivors: A Systematic Review. International Journal of Environmental Research and Public Health, 19(19), 12704.
- Breuer, N., Sender, A., Daneck, L., Mentschke, L., Leuteritz, K., Friedrich, M., ... & Geue, K. (2017). How do young adults with cancer perceive social support? A qualitative study. Journal of Psychosocial Oncology, 35(3), 292-308.
METHODS
1. How the recruitment was conducted? Please, add more details
2. Which researchers conducted each research study's phase?
RESULTS
1. I am not sure about the relevance of the presented findings. Did you conduct just descriptive analyses obtaining scores in percentage? Only distribution among countries revealed significant results in terms of p-values? Please, make your result more consistent in promoting novelty
DISCUSSION AND CONCLUSION
1. Please, add practical implications
Round 2
Reviewer 1 Report
Dear Authors,
Thank you for the opportunity to review this paper again. Thank you for taking into account my few comments and for all the extensive explanations too. Unfortunately, most of the key comments were not included in the publication, though I understand the explanation to some extent. I maintain my opinion that my main comments would affect the scientific quality of the publication. Nevertheless, if the authors are unable to obtain certain data (e.g. the number of people from the population with certain socio-economic characteristics), the presented data have considerable limitations, also in terms of the place of publication.
Kind regards
reviewer
Author Response
Thank you for your comments and we revised the manuscript accordingly

Reviewer 4 Report
No other comments to add
Author Response
Thank you for your time to provide your comments and responses